# Ric-8A, a G protein chaperone with nucleotide exchange activity induces long-range secondary structure changes in Gα

Ravi Kant[1†], Baisen Zeng[2†], Celestine J Thomas[2‡], Brian Bothner[1*], Stephen R Sprang[2*]

[1]Department of Chemistry and Biochemistry, Montana State University, Bozeman, United States; [2]Center for Biomolecular Structure and Dynamics, The University of Montana, Missoula, United States

**Abstract** Cytosolic Ric-8A has guanine nucleotide exchange factor (GEF) activity and is a chaperone for several classes of heterotrimeric G protein α subunits in vertebrates. Using Hydrogen-Deuterium Exchange-Mass Spectrometry (HDX-MS) we show that Ric-8A disrupts the secondary structure of the Gα Ras-like domain that girds the guanine nucleotide-binding site, and destabilizes the interface between the Gαi1 Ras and helical domains, allowing domain separation and nucleotide release. These changes are largely reversed upon binding GTP and dissociation of Ric-8A. HDX-MS identifies a potential Gα interaction site in Ric-8A. Alanine scanning reveals residues crucial for GEF activity within that sequence. HDX confirms that, like G protein-coupled receptors (GPCRs), Ric-8A binds the C-terminus of Gα. In contrast to GPCRs, Ric-8A interacts with Switches I and II of Gα and possibly at the Gα domain interface. These extensive interactions provide both allosteric and direct catalysis of GDP unbinding and release and GTP binding.

*For correspondence: bbothner@
chemistry.montana.edu (BB);
stephen.sprang@umontana.edu
(SRS)

[†]These authors contributed
equally to this work

**Present address:** [‡]Regeneron
Pharmaceutical, Inc., New York,
United States

**Competing interests:** The
authors declare that no
competing interests exist.

**Reviewing editor:** John Kuriyan,
Howard Hughes Medical
Institute, University of California,
Berkeley, United States

## Introduction

Resistance to Inhibitors of Cholinesterase 8A (Ric-8A) is a 59.7 KDa protein that catalyzes the release of GDP from i, q and 13-classes of heterotrimeric G protein α subunits (Gα) in vitro (*Tall et al., 2003*). In that capacity, Ric-8A is the best characterized of the family of cytosolic proteins with guanine nucleotide exchange (GEF) activity at Gα subunits. Unlike G protein-coupled receptors, which act upon heterotrimeric complexes of Gα and G protein beta-gamma heterodimers (Gβγ), Ric-8A acts exclusively at GDP-bound Gα. GEF activity is unidirectional; Ric-8A does not interact with Gα•GTP, nor does the Gα•Ric-8A complex dissociate in the presence of GDP (*Tall et al., 2003*). Ric-8A is structurally unrelated to GPCRs. The circular dichroic spectrum of mammalian Ric-8A indicates that it is a predominantly alpha-helical protein (*Figueroa et al., 2009*; *Thomas et al., 2011*) and amino acid sequence analysis suggests that Ric-8A adopts an armadillo-repeat fold similar to that of β-catenin and its structural analogs (*Figueroa et al., 2009*). Among the latter are nuclear import proteins that are regulated by the Ras family G protein Ran (*Vetter et al., 1999*), and smgGDS, a GEF for certain Rho-family G proteins (*Hamel et al., 2011*).

In *C. elegans*, *Drosophila* and mouse, Ric-8A orthologs (generally, Ric-8) have been identified as regulators of synaptic transmission and of mitotic spindle movement in asymmetric cell division (*Afshar et al., 2004*; *Couwenbergs et al., 2004*; *David et al., 2005*; *Wang et al., 2005*; *Miller and Rand, 2000*). In the latter role, Ric-8 orthologs may function as components of G Protein Regulator (GPR/Goloco)-signaling complexes with Gα (*Woodard et al., 2010*; *Oner et al., 2013*; *Thomas et al., 2008*). Roles in amplification of G protein signaling in chemotaxis by *Dictyostelium* (*Kataria et al., 2013*) and synaptic transmission in *C. elegans* (*Miller et al., 2000*; *Reynolds et al.,*

*2005*) have also been described. There is no direct evidence that Ric-8A is a physiological GEF. However the observation of Ric-8A-enhanced Gβγ-dependent signaling after peptide-induced disruption of G protein heterotrimers is consistent with in vivo GEF activity (*Malik et al., 2005*), as is its presence in the mitotic spindle-associated complexes described above.

Recent studies have provided evidence that Ric-8 facilitates the biogenesis of active Gα proteins in eukaryotic cells and their localization to intracellular membranes and the cell cortex (*Gabay et al., 2011*; *Hampoelz et al., 2005*; *Wang et al., 2011*). Ric-8 orthologs function as molecular chaperones for Gα (*Chan et al., 2013*). By inhibiting their ubiquitination, Ric-8 also blocks the degradation of Gα subunits (*Nagai et al., 2010*; *Chishiki et al., 2013*). Whether Ric-8 isoforms have dual physiological functions as both chaperones and GEFs has not been resolved (*Blumer and Lanier, 2014*; *Tall, 2013*; *Hinrichs et al., 2012*). In any case, Ric-8A is essential for mammalian development. Ric-8A (-/-) mutant mouse embryos cannot progress beyond the initial stages of gastrulation and die during early embryogenesis (*Tõnissoo et al., 2010*).

Gα subunits are members of the Ras superfamily of regulatory GTP binding proteins (*Gilman, 1987*; *Sprang, 1997*). Characteristic of the superfamily, Gα bear polypeptide segments called Switch I and Switch II that flank the GTP binding site. These segments are observed to undergo conformational transitions upon GTP hydrolysis that are typically coupled to the loss of affinity for effectors and an increase in affinity for Gβγ (*Lambright et al., 1994*; *Mixon et al., 1995*). Other characteristic structural features are a P-loop that encompasses the α and β phosphates of the guanine nucleotide and two loop segments that form conserved contacts with the purine ring (*Noel et al., 1993*; *Coleman et al., 1994*). Gα subunits are also distinctive among Ras superfamily members in the insertion of a ~110 residue alpha helical domain at the N-terminus of Switch I. The helical domain blocks egress of nucleotide from the Ras domain. G Protein-Coupled Receptors (GPCR), as exemplified by the complex of the β2 adrenergic receptor (β2AR) with the Gs heterotrimer, grasp the C-terminus of Gα and the N-terminus of its β1 strand. These interactions are proposed to cause structural perturbations that disrupt the P-loop and purine binding site, and release contacts between the Ras and alpha helical domains, thereby liberating GDP and providing an entry path for GTP (*Rasmussen et al., 2011*).

Ric-8A forms a Michaelis complex with Gα•GDP which, in the absence of GTP, releases GDP and forms a stable Gα:Ric-8A intermediate. In the presence of GTP, this complex undergoes an exchange reaction, generating free Ric-8A and Gα•GTP (*Tall et al., 2003*; *Thomas et al., 2008*). Using several biophysical approaches, we demonstrated that Gαi1 is conformationally heterogeneous in the Gαi1:Ric-8A complex (*Thomas et al., 2011*). We observed severe attenuation of amide $^{15}N$-$^1H$ peaks in the TROSY-HSQC spectrum of Gαi1, and a doubling in the number of protons that become accessible to proton-deuterium exchange relative to Gαi1•GDP. Double electron-electron resonance (DEER) experiments with nitroxide spin-labeled Gαi1 showed extensive broadening in the spectrum of inter-probe distances that accompany displacement of the alpha helical and Ras-like domains of the G protein (*Van Eps et al., 2015*). In stabilizing the nucleotide-free state, Ric-8A induces or accommodates large changes in the global structure and dynamic behavior of Gαi1.

Here, we report experiments using time-resolved HDX-MS to observe, at a resolution of 5–15 residues, the response of Gαi1 and Ric-8A secondary structures to formation of the nucleotide-free Gαi1:Ric-8A intermediate. At near neutral pH, in the seconds-to-hours time regime, continuous exchange experiments report HDX from peptide amide groups. HDX is suppressed by involvement of backbone amide hydrogen atoms in hydrogen bonds, but enhanced when such interactions are broken and amide groups become accessible to solvent. Further, protein 'breathing' motions that expose amide protons to solvent result in a steady increase in HDX over time (*Drozdetskiy et al., 2015*; *Englander and Kallenbach, 1983*). Protein-protein and protein-ligand interactions may induce changes in amide solvent accessibility, protein flexibility and hydrogen bonding, and these are manifest in changes in HDX processes. Here, we show that in the complex with Ric-8A, the peptide amide hydrogen atoms of nucleotide-free Gαi1 become accessible to exchange, not only in amino acids that contact the nucleotide, but also in the scaffold of secondary structure elements that encompass and support the nucleotide-binding site. The direction of these changes is essentially the reverse of those observed in the transition from Gαi1•GDP to Gαi1 bound to $Mg^{2+}$ and the slowly hydrolysable GTP analog GTPγS, although greater in magnitude. Moreover, HDX measurements reveal a potential Gα binding site in Ric-8A, which we show by mutagenesis, contributes substantially to Ric-8A GEF activity.

## Results and discussion

### Acquisition and analysis of HD-MX data

We analyzed the amide hydrogen exchange properties of rat Gαi1 in complex with rat Ric-8A, the stable intermediate for Ric-8A-catalyzed nucleotide exchange in which the nucleotide-binding site of Gαi1 is empty. Free Ric-8A and Gαi1•GDP are the reference states to which both components of the Gαi1:Ric-8A complex are compared. We also measured the HDX properties of Gαi1•Mg$^{2+}$GTPγS (henceforth Gαi1•GTPγS) in which Gαi1 is activated for effector engagement. Gαi1•GDP is the reference state for the latter as well. In the experiments described here, we used non-myristoylated rat Gαi1 and rat (1-491) Ric-8A, in which the 29 C-terminal residues of the full-length protein are absent. This C-terminally truncated Ric-8A exhibits roughly 10% higher GEF activity than the native protein (*Thomas et al., 2008*) and is henceforth referred to as Ric-8A. The Gαi1:Ric-8A complex remains intact and active in GTPγS uptake throughout the course of the of the HDX experiment (*Figure 1— figure supplement 1*).

We measured HDX from Gαi1•GDP, Gαi1•GTPγS, Ric-8A and Gαi1:Ric-8A at successive time points ranging from 30 s to 5 hr following dilution of proteins or protein complexes with deuterated buffer. These were then subjected to pepsin digestion at pH 2.5°C and 0°C to minimize back-exchange, yielding peptide fragments ranging in length from four to ~20 amino acid residues. In aggregate, overlapping peptide fragments recovered from pepsinolysis represent the entire amino acid sequence of Gαi1 and 87% of that of Ric-8A (*Figures 1–3*). The average and maximum number of peptides that overlap with any single residue is 2.5 and 6, respectively, for Gαi1 and 1.7 and 6, respectively for Ric-8A. Time-dependent HDX from Gαi1•GDP, Gαi1•GTPγS, Gαi1:Ric-8A and Ric-8A, are shown as heat maps in *Figures 1–3* and the change in deuteration for all peptides with incubation is reported in *Figure 1–source data 1*, *Figure 2–source data 1*, *Figure 3–source data 1*. The absolute extent of exchange from corresponding peptides derived from different preparations of Gαi1 is not identical. However the same preparation of Gαi1 was used to measure the difference in the extent of HDX for the pair Gαi1•GDP and Gαi1:Ric-8A, and for the pair Gαi1•GDP, Gαi1•GTPγS. Likewise, the same preparation of Ric-8A was used to measure differences in HDX properties of free Ric-8A and its complex with nucleotide-free Gαi1. These differences are mapped onto the three-dimensional structures of Gαi1 determined by X-ray crystallography (*Figures 4–6* and *Video 1*) and a representative computationally-derived (*Bradley et al., 2005*) model of the tertiary structure of Ric-8A obtained by use of the Robetta server (*Kim et al., 2004*) (*Figure 7a*). The overall pattern of HDX from Gαi1 secondary structures in the GTPγS-bound state is similar to that reported earlier for the GDP•Mg$^{2+}$AlF complex of Gαi1 published earlier (*Preininger et al., 2012*).

### Secondary structure plasticity of Gαi1 in the Gαi1:Ric-8A complex

Comparison of the fast HDX behavior of nucleotide-free Gαi1 bound to Ric-8A with that of Gαi1•GDP (*Figure 4a*) reveals changes in solvent accessibility throughout the structure of Gαi1 during the first 60 s of exposure to D$_2$O. Ric-8A binding and GDP release result in structural changes manifested by up to a 40% increase in deuteration (deprotection) of some peptides and a decrease in deuteration exceeding 25% (protection) in others. In contrast, Ric-8A binding does not induce striking changes in slow protein dynamics (*Figure 1—source data 1*) as measured by the slope of HDX with time. Most of the peptide segments that directly contact the nucleotide-binding scaffold of Gαi1 are deprotected in the Ric-8A-bound state. The P-loop, which enfolds the α and β phosphates of bound guanine nucleotides, and the β5-αG loop, which harbors the NPXY motif that determines guanine- specificity at the purine binding sub-site, show high levels of deprotection (*Figures 4a* and *5*, *Video 1*). Backbone amide deprotection extends to the α-helical (αG and a segment of α4) and β-sheet scaffolds (β1, β4 and β5) that support these nucleotide-contacting regions (*Figure 5b*, *Video 1*). Secondary structure elements αG, β4 and β5 in particular support a hydrophobic cluster that supports the purine binding site (*Figure 5c*, *Video 1*). Increased deuteration of these segments indicates that this main-chain hydrogen bonding network is destablized in the Gαi1:Ric-8A complex. van der Waals interactions within the cluster could in turn be disrupted, resulting in the loss of tertiary structural integrity of the nucleotide-binding site and the surrounding structure. In contrast, the segment extending from the C-terminus of α4 though β6 is protected by Ric-8A, as is the C-terminus of α5 (see below).

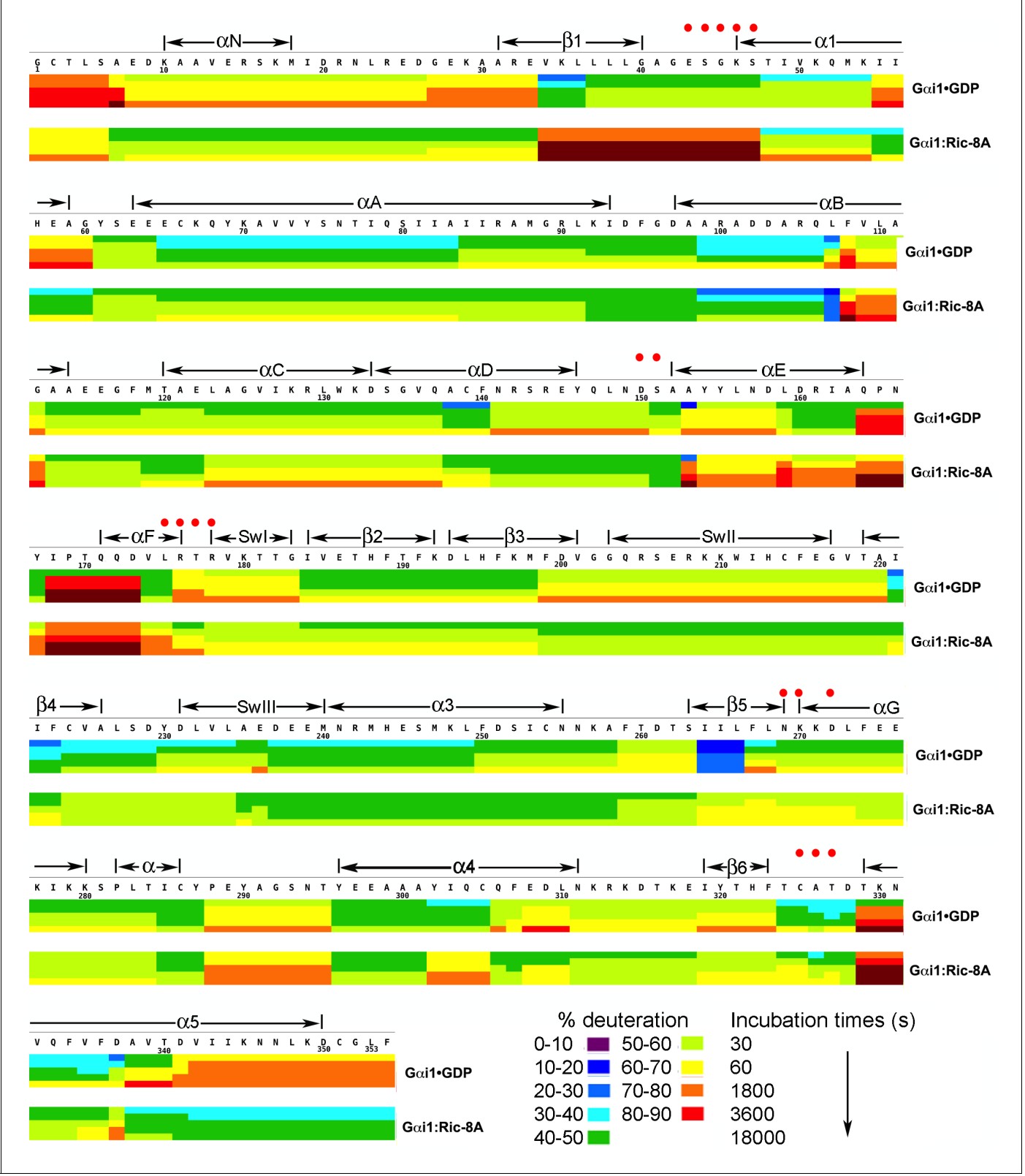

**Figure 1.** Kinetics of Hydrogen-Deuterium exchange from Gαi1 in complexes with GDP and Ric-8A. HDX at successive time points are represented by sets of horizontal bars, progressing from top (30 s) to bottom (5 hr), mapped on the amino acid sequence of Gαi1 for Gαi1•GDP (upper set) and Gαi1:Ric-8A (lower set). Color coding (see key) indicates fraction (percent) of total amide hydrogen atoms exchanged per peptide at each time point.

*Figure 1 continued on next page*

*Figure 1 continued*

Location of secondary structure elements is shown above the amino acid sequence. Red dots indicate residues that form non-covalent interactions with GDP.

The following source data and figure supplement are available for figure 1:

**Source data 1.** Percent deuteration of peptides derived from Gαi1•GDP and Gαi1:Ric-8A.

**Figure supplement 1.** The Gαi1:Ric-8A complex is stable over the period in which the HDX experiments were performed.

Secondary structure elements at the Ras domain-helical domain interface are destabilized in the Gαi1:Ric-8A complex. Conserved residues in αF, α5 and, to a lesser extent, α1 all show increased deuteration (*Figure 6*, *Video 1*). These residues contribute to a network of interactions that link the domain interface and the guanine nucleotide-binding site. Structural elements β6-α5, αF, the P-loop and the succeeding α1 helix are all involved. It is likely that destabilization of these interactions promotes nucleotide unbinding, domain separation (*Van Eps et al., 2015*) and nucleotide release.

In contrast, the αD-αE loop in the helical domain becomes moderately protected in Gαi1:Ric-8A (*Figure 4a*), even as the juxtaposed αG and Switch III in the Ras domain are strongly deprotected. The αD-αE loop, together with other features of the helical domain, including αA, may be shielded from solvent by interactions with Ric-8A.

## HDX protection points to a potential Ric-8A-interaction surface of Gαi1

Switch I (~residues 176–181) and to a greater extent, Switch II (~residues 203–215), show modest protection from HDX in the Ric-8A complex (*Figure 4a* and *Figure 5a*, *Video 1*). Other evidence implicates these regions as possible loci of direct interaction between Gαi1 and Ric-8A. Specifically, EPR studies show that Gαi1 residues R209 and K180 in Switch I and Switch II, respectively, become partially immobilized upon binding to Ric-8A (*Van Eps et al., 2015*). The possibility that Switch II is a Ric-8A binding site is also suggested by the observation that a chimera of Gαs that harbors the Switch II sequence of Gαi1 is not protected from ubiquitination by Ric-8B, a Gαs-specific ortholog of Ric-8A (*Nagai et al., 2010*). The C-terminus of Gαi1, including residues of α5, is strongly protected from HDX by Ric-8A (*Figure 4*). This finding is consistent with other studies that implicate the C-terminus of Gαi1 as a Ric-8A interaction site, including the observation that a peptide encompassing this sequence is a competitive inhibitor of Gαi1 for Ric-8A GEF activity (*Papasergi et al., 2015*; *Thomas et al., 2011*). Also protected is the amino terminus of Gαi1, which packs against the C-terminus in crystal structures of Gαi1•GDP that crystallize in space group I4 (*Mixon et al., 1995*). Nevertheless, an N-terminal truncation mutant of Gαi1 remains a Ric-8A substrate (*Thomas et al., 2011*). From the HDX protection data, we infer a broad interaction surface for Gαi1 that encompasses structural elements of the helical domain near its interface with the Ras domain, extends to Switch I and Switch II, and includes the C-terminus, which may make the most intimate contacts with Ric-8A (*Figure 4a*, *Video 1*).

## GDP → GTP exchange reverses Ric-8A – induced conformational changes

Changes in HD protection that occur upon exchange of GDP for GTPγS are, for the majority of peptides analyzed, less pronounced than those observed when Ric-8A displaces GDP from Gαi1 (*Figure 4b*). In the P-loop and α1 at the nucleotide-binding site, changes in the magnitude of HDX are the opposite of those observed in the transition from Gα•GDP to Gα•Ric-8A. These latter segments become moderately protected, consistent with structural rigidification induced by nucleotide triphosphate binding. Modest protection extends also to the β4 – Switch III – α3 turn sequence, and αD-αF loop that flanks Switch III in polar inter-domain contacts. Notably, Switch III and α3 are consensus effector binding sites of GTP-activated Gα (*Sprang et al., 2007*) and protection from HDX could reflect a structural stiffening of these elements in the GTP-bound state. The pattern of changes in HDX protection upon transition between GDP and GTPγS-bound states roughly parallel the changes in amide chemical shifts observed in the HSQC spectra of Gαi1 (*Goricanec et al., 2016*).

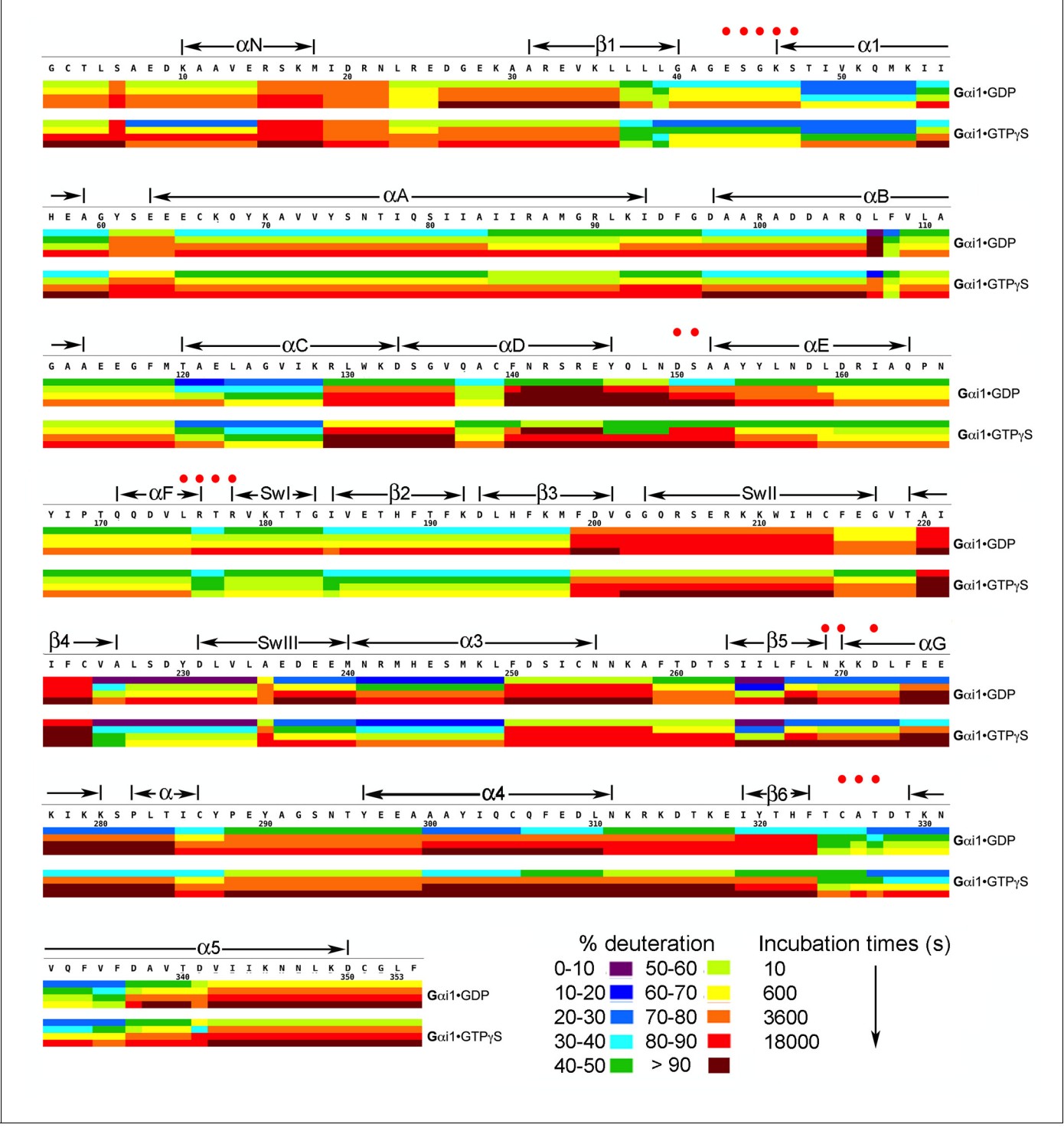

**Figure 2.** Kinetics of Hydrogen-Deuterium exchange from Gαi1 in complexes with GDP and GTPγS. Image elements are as described in the legend to *Figure 1*.

The following source data is available for figure 2:

**Source data 1.** Percent deuteration of peptides derived from Gαi1•GDP and Gαi1•GTPγS after incubation in deuterated buffer.

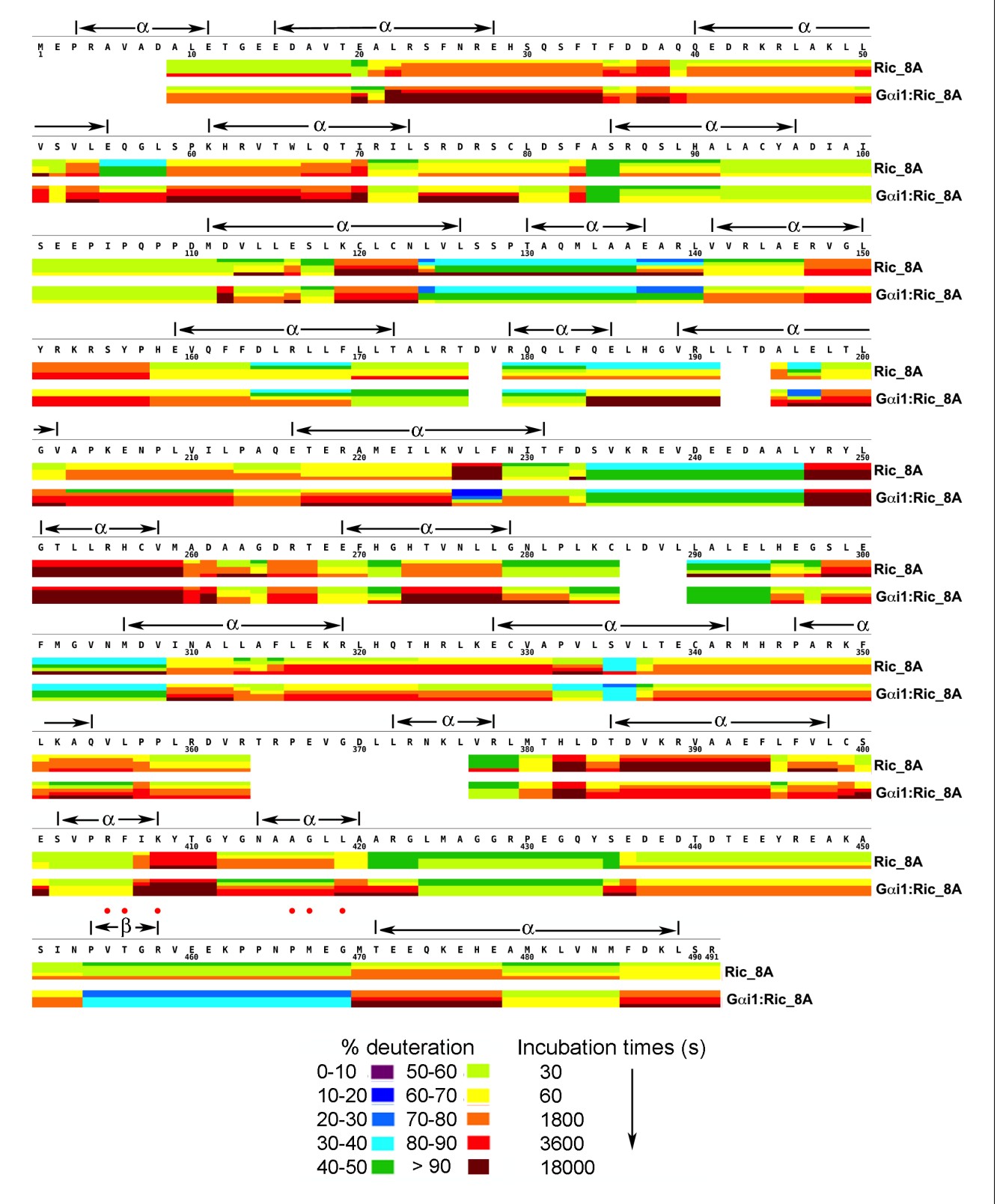

**Figure 3.** Kinetics of Hydrogen-Deuterium exchange from Ric-8A, free and in the complex with Gαi1. Location of predicted secondary structure elements is shown above the amino acid sequence. Red dots indicate residues which, upon substitution with alanine, result in significant impairment of GEF activity. Image elements are otherwise as described in the legend to *Figure 1*.

*Figure 3 continued on next page*

*Figure 3 continued*

The following source data is available for figure 3:

**Source data 1.** Percent deuteration of peptides derived from Ric-8A and Gαi1:Ric-8A after incubation in deuterated buffer.

A subset of Gαi1 peptides show similar changes in HDX in response to binding of Ric-8A or GTPγS. The C-terminus of β3 and Switch II are protected from HDX by GTPγS binding. Residues in this segment, particularly the peptide amides of 201–203, engage the γ phosphate of GTP and Asp 200 is hydrogen bonded to a water ligand of the $Mg^{2+}$ co-factor (*Coleman et al., 1994*; *Coleman and Sprang, 1999*). Switch II itself, which is poorly ordered in the GDP-bound state, adopts a more ordered helical conformation and packs against α3 and the β1-β3 scaffold of the Ras domain (*Mixon et al., 1995*). The moderate protection of Switch I in the GTPγS-bound state may reflect the stabilization of this element though interactions of R178 and T181 with the nucleotide triphosphate and the metal cofactor. Thus, relative to Gαi1•GDP, switches I and II experience increased protection from HDX in both the Ric-8A and GTPγS complexes, although by different mechanisms: direct interaction with Ric-8A in the first instance, and structural rigidification upon GTPγS•$Mg^{2+}$ binding in the second.

The transition from GDP to GTP-bound states does not affect HDX at sequences (β5 - αG and β6 - α5) that contact the guanine base of GTP. This is consistent with crystal structures, which show

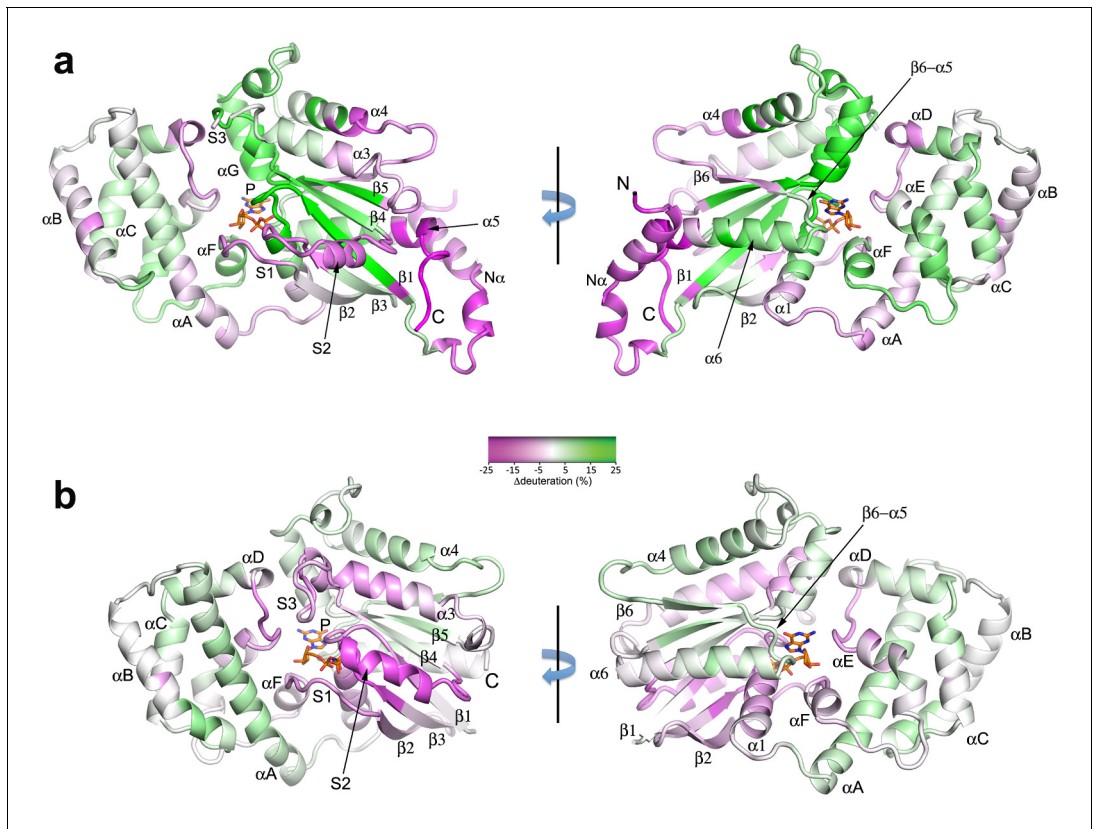

**Figure 4.** Binding of Ric-8A to Gαi1, and nucleotide exchange induce widespread changes in Gα secondary structure. (a) in the transition from Gαi1•GDP to Gαi1:Ric-8A-bound states, and (b) Gαi1•GDP to GTPγS complexes, as revealed by HDX-MS. The color scheme represents difference in per-residue deuteration relative to Gαi1•GDP mapped onto the structures of Gα•GDP (A, PDB 1GGD) or Gαi1•GTPγS (PDB 1GIT) Residues that undergo an increase in deuteration of 25% or greater are colored green (deprotection), and those that experience a decrease in deuteration of 25% or less are colored magenta (protection). Intermediate degrees of deuteration changes are colored according to the color key. Secondary structure elements are labeled; Switch I, Switch II and Switch III are labeled as S1, S2 and S3, respectively. Bound nucleotide, which is not present in the Gαi1:Ric-8A complex, is shown as a stick figure.

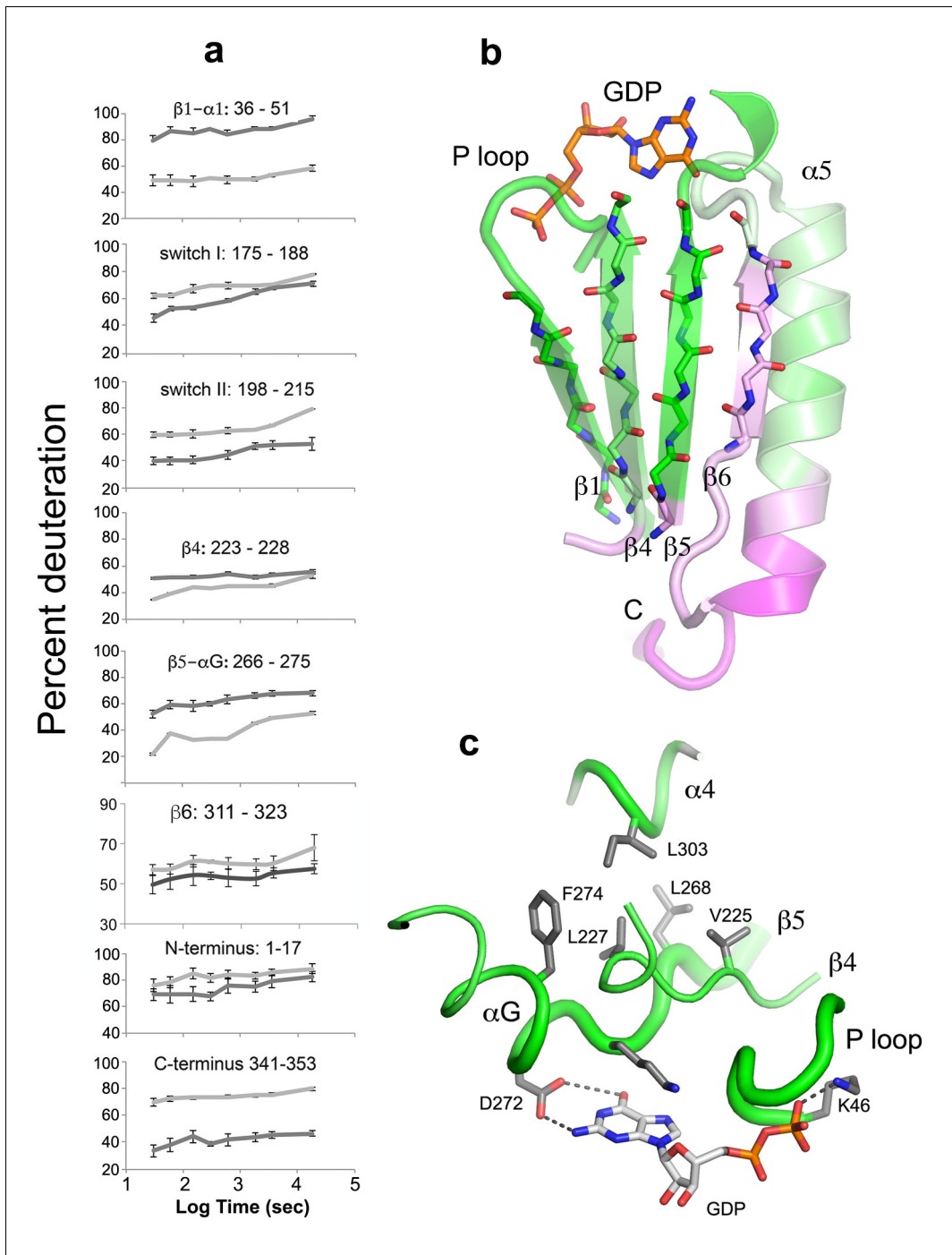

**Figure 5.** The beta sheet scaffold and nucleotide-binding site of Gαi1 are deprotected in the nucleotide-free complex with Ric-8A. (a) kinetics of HDX at selected Gαi1 peptides in Gαi1•GDP, *light gray*, and Gαi1•Ric-8A, *dark gray*; error bars represent the standard deviation computed for three technical replicates (see Materials and methods) (b) section of the beta sheet and α5 in the Ras domain that abut the nucleotide-binding site, colored according to the extent of HDX, using the coloring scheme used in *Figure 4*; (c) elements of the nucleotide-binding site featuring the hydrophobic cluster between αG, β4, β5 and α4. Tube diameter is proportional to deuteration. Side chains and GDP are shown as stick figures: carbon, *gray*, oxygen, red and nitrogen, blue.

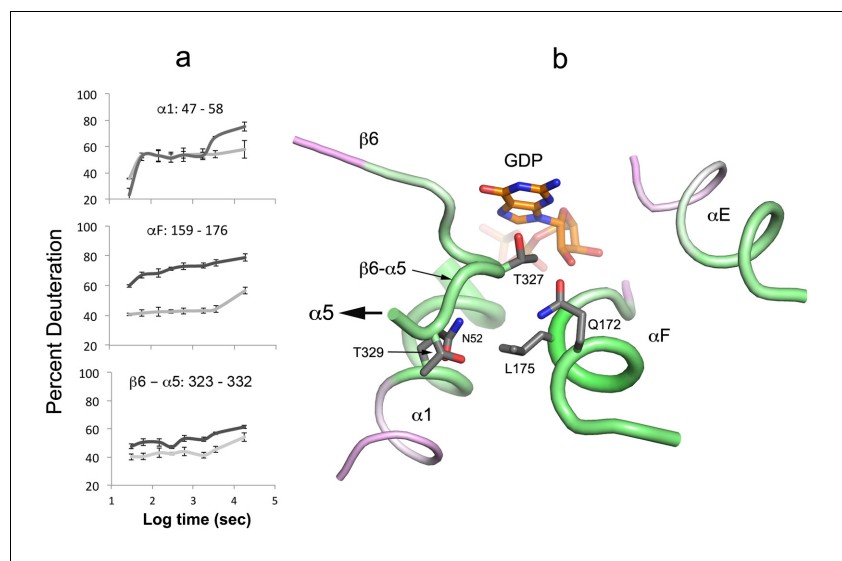

**Figure 6.** Destabilization of secondary structure at the Ras-domain:helical domain interface. (**a**) HDX kinetics of interdomain secondary structure elements. (**b**) side chains of conserved residues in human Gα isoforms.

residues in these two segments to be well ordered in both GTP and GTP-analog bound complexes of Gαi1 and transducin (*Mixon et al., 1995*; *Lambright et al., 1996*). Several structural elements show mild deprotection in the GTP-bound state. These include αA in the helical domain, and helical segments against which it packs, as well as residues at the N-terminus of β4 and in the α4-β6 purine-binding loop in the Ras domain. The structural origin of these changes is not apparent.

## HD protection identifies potential Gαi1 binding site on Ric-8A

The three dimensional structure of Ric-8A has not been determined, nor does it have significant amino acid sequence identity to members of any protein families represented in the Protein Data Bank. We therefore used the Robetta web server (*Kim et al., 2004*) to execute an automated sequence of Rosetta structure prediction tools (*Bradley et al., 2005*) that yielded five all-atom models that ranked highest with respect to specific stereochemical criteria. In agreement with earlier sequence analysis studies (*Figueroa et al., 2009*), Ric-8A is predicted to adopt an armadillo fold, similar to that of β-catenin and nuclear import proteins. The Ric-8A models differed mainly with respect to the packing interactions between neighboring helical segments within the ~130 residue

C-terminal region and the spatial relationship of the latter to the N-terminal 400 residues, suggesting that the two regions constitute distinct structural domains. Throughout its primary structure, Ric-8A is highly susceptible to HDX (*Figure 3*), with nearly half of the observed peptide segments incorporating deuterium at 60% of exchangeable sites. Upon binding to Gαi1, Ric-8A undergoes changes in accessibility to HDX throughout its amino acid sequence (*Figure 7a*). Many Ric-8A-derived peptides show 5–15% changes in deuteration relative to free Ric-8A at successive α/α repeats, suggestive of distributive conformational changes.

Notably, an extended sequence corresponding to the structural repeats in Ric-8A, from residue 425 to the C-terminus of the construct (residue 491) exhibit an 10–15% increase in HDX

Supplemental Movie 1

"Ric-8A, a G protein chaperone with nucleotide exchange activity induces long-range secondary structure changes in Gα"

Authored by: R███████, Baisen Zeng, Celestine J. Thomas, Brian Bothner, and Stephen R. Sprang

Movie by: Angela Patterson

**Video 1.** HDX changes in Gαi1•GDP upon binding of Ric-8A.

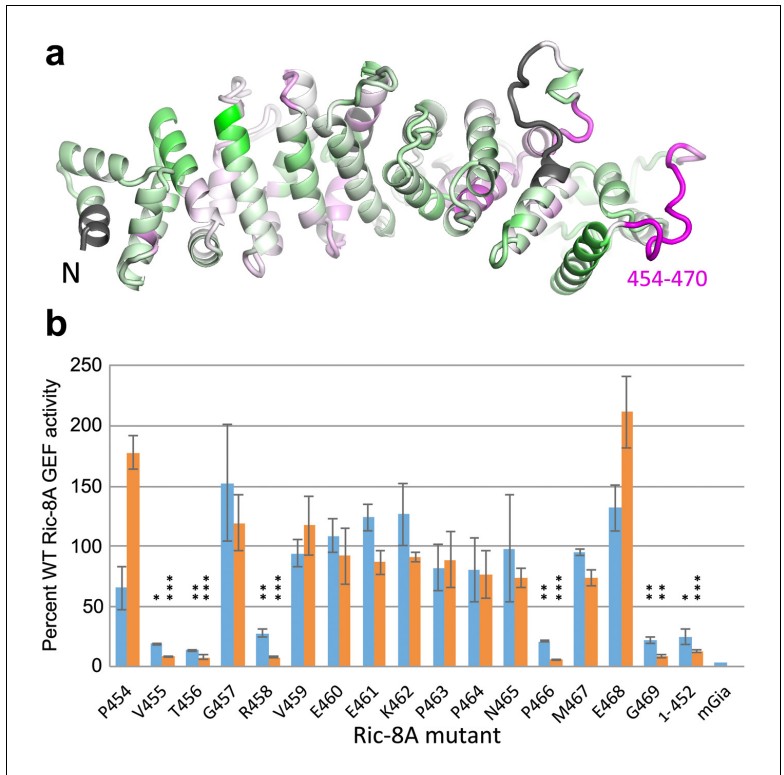

**Figure 7.** HDX identifies a possible Gαi1 binding site in Ric-8A. (a) amplitude of HDX mapped on a model of the tertiary structure of Ric-8A, using the coloring scheme adopted in *Figure 4*; (b) GEF activity of Ric-8A mutants expressed as percent of WT Ric-8A activity. Each Ric-8A variant is mutated to alanine at one of the residues within the sequence 454–470 as indicated. GEF activity for the (1-452)Ric-8A truncation mutant is also shown. Relative GEF activity represented by the blue bars is measured as the initial velocity of GDP to GTPγS exchange at GDP-bound Gαi1 (2 μM), upon addition of GTPγS (10 μM) and Ric-8A (2 μM) [WT rate, 3.84 (1.45) μM/min (sample standard deviation in parentheses), n = 8] *Orange* bars represent the relative rate of GTPγS binding to nucleotide-free Gαi1:Ric-8A (2 μM) [WT rate, 4.66 (1.38), n = 8]. Error bars show standard deviation of mutant/WT GEF activity ratios based on three independent measurements. The bar labeled mGiα quantifies the intrinsic exchange rate of myristoylated Gαi1 [0.08 (0.02)] $min^{-1}$. Asterisks above bars indicate significance (two-tailed Student's t test) of the difference in the respective mutant Ric-8A-catalyzed rate to that of WT Ric-8A: *$p<0.01$; **$p<0.005$; ***$p<0.001$. For cases in which the difference between these rates is corresponds to $p>0.01$), no asterisk is shown.

relative to free Ric-8A. This pattern is interrupted by a highly protected (25% decrease in deuteration) peptide comprising residues 454–470, which is rich in proline and acidic residues. Based on sequence analysis of more than 250 Ric-8A homologs using Jpred4 (*Drozdetskiy et al., 2015*), this polypeptide is predicted to be largely unstructured.

The substantial protection afforded residues 454–470 in the Gαi1:Ric-8A complex implicates this segment to be a possible Gαi1 binding site. To test this hypothesis, we conducted an alanine scan of this region, and assayed the GEF activity of each alanine mutant using a tryptophan fluorescence assay that reports the exchange of GDP for GTPγS at Gαi1 (*Higashijima et al., 1987a*). For each mutant, we measured the initial velocity of nucleotide exchange from Gαi1•GDP upon addition of GTPγS and Ric-8A (blue bars in *Figure 7b*, *Table 1*). We also measured the initial rate of incorporation of GTPγS into Ric-8A-bound Gαi1, which results in dissociation of Ric-8A from the Gαi1•GTPγS (orange bars in *Figure 7b*, *Table 1*).

Alanine scanning revealed several 'hotspots' for GEF activity, clustered at the N- and C-termini of the 454–470 sequence. Experiments were performed with myristoylated Gαi1 harboring a hexahistidine tag in the helical domain (*Mumby and Linder, 1994*). Mutation of any of the five hotspot residues (V455, T456, R458, P466 and G469) resulted in a diminution in the initial rate of GTPγS incorporation to 12–20% that of WT Ric-8A. Using differential scanning fluorimetry, we found that

**Table 1.** Initial velocities of Ric-8A mutant guanine nucleotide exchange activity. Assays were conducted as described in Materials and methods section of the Main text. For measurement of $v_1$, reaction buffer (50 M HEPES, pH 8.0, 150 M NaCl, 10 M MgCl$_2$, and 1 M TCEP) at 25°C contained 2 μM Gαi1•GDP, 2 μM Ric-8A and 10 μM GTPγS (initial concentration); for measurement of $v_2$, 2 μM Gαi1•GDP, 2 μM Ric-8A were incubated for 5 min before addition of 10 μM GTPγS.

| Ric-8A mutant | $v_1$* (μM/min) | $v_2$ (μM/min) |
| --- | --- | --- |
| WT | 3.84 (1.45) | 4.66 (1.38) |
| P454A | 2.88 (0.80) | 5.39 (0.43) |
| V455A | 0.82 (0.05) | 0.25 (0.01) |
| T456A | 0.59 (0.04) | 0.24 (0.06) |
| G457A | 3.47 (1.11) | 6.02 (1.19) |
| R458A | 0.63 (0.07) | 0.37 (0.03 |
| V459A | 5.65 (0.68) | 6.85 (1.44) |
| E460A | 6.54 (0.83) | 5.36 (1.39) |
| E461A | 7.45 (0.65) | 5.06 (0.59) |
| K462A | 4.46 (0.92) | 5.15 (0.24) |
| P463A | 2.90 (0.69) | 4.78 (1.26) |
| P464A | 2.83 (0.94) | 4.73 (1.21) |
| N465A | 2.25 (1.03) | 4.56 (0.51) |
| P466A | 0.47 (0.01) | 0.34 (0.02) |
| M467A | 2.17 (0.06) | 4.57 (0.44) |
| E468A | 3.82 (0.55) | 6.23 (0.88) |
| G469A | 0.63 (0.08) | 0.26 (0.05) |
| (1-452)Ric-8A | 0.86 (0.22) | 0.39 (0.03) |

* $v_1$ for the reaction
Gαi1•GDP + GTPγS + Ric-8A → Gαi1•GTP + Ric-8A + GDP
$v_2$ for the reaction
Gαi1•Ric-8A + GTPγS → Gαi1•GTPγS + Ric-8A
Values in parentheses are the standard deviation for three independent experiments.

most of the mutants were slightly destabilized, with thermal unfolding transitions (Tm) ranging from 38°C to 47°C, relative to WT Ric-8A, which transitions at 47°C (*Figure 8*). There is no apparent correlation between thermal melting temperature and GEF activity for these mutants. Comparable loss of GEF activity is achieved by C-terminal truncation of Ric-8A at residue 452, which entirely eliminates the protected sequence. Alanine mutants at residues 454 and 468 exhibited significantly slower rates for the overall exchange reaction than for exchange from the Gαi1:Ric-8A intermediate (p<0.001, P454A and p<0.02, R458A). Mutation of these residues appears to facilitate binding of GTPγS to, or release of Ric-8A from, the Gαi1:Ric-8A intermediate. We propose that residues at the N- and C-termini of the protected sequence in Ric-8A interact directly or through an allosteric interaction with the empty nucleotide-binding site of Gαi1 to increase its affinity for GTP.

## Ric-8A and GPCRs engage Gα by different mechanisms

The HDX results presented here suggest that GPCRs and Ric-8A adopt different strategies to induce roughly similar structural transitions in Gα that facilitate nucleotide exchange. This is not unexpected, since the two GEFs are structurally unrelated and act on different states of Gα: GPCRs on Gα•GDP bound to Gβγ, and Ric-8A on free Gα•GDP. Biochemical (*Oldham and Hamm, 2008*), structural (*Van Eps et al., 2015*; *Westfield et al., 2011*; *Chung et al., 2011*), computational (*Dror et al., 2015*) and conservation-based modeling studies

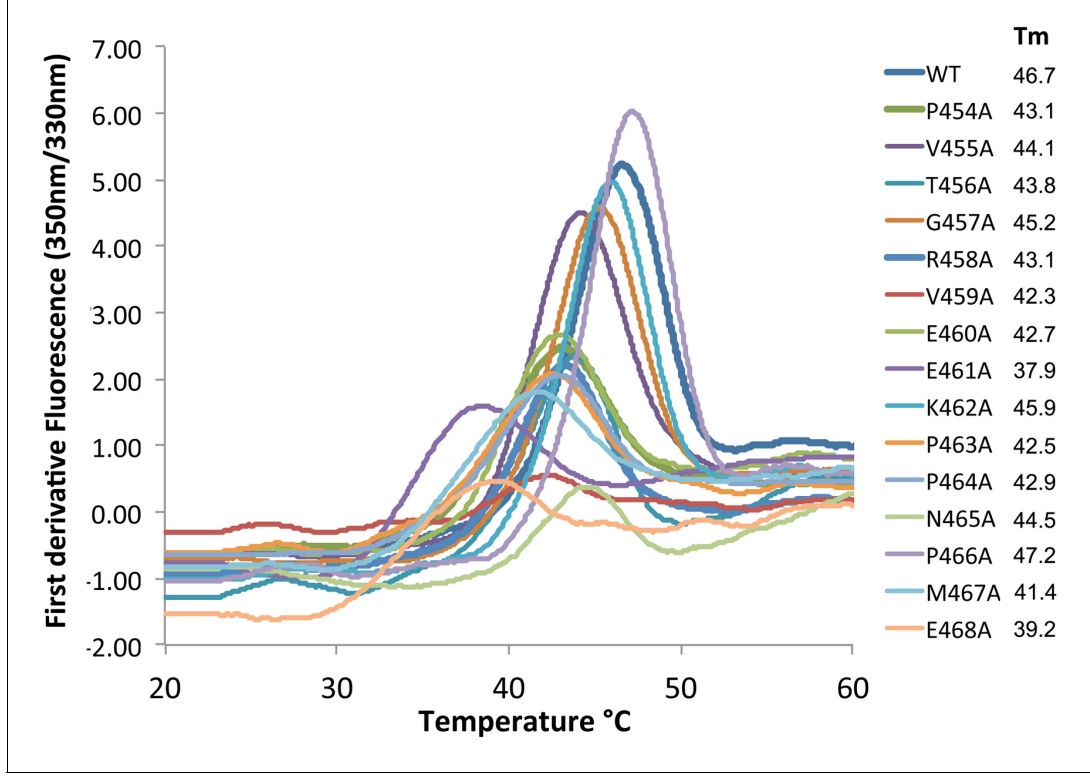

**Figure 8.** Differential scanning fluorimetry of Ric-8A and its mutants. Samples (10 μl) of wild type Ric-8A(1–491) and mutants thereof (~1 mg/ml) in 50 mM HEPES, pH 8.0, 150 mM NaCl and 1 mM TCEP were dispensed into glass capillaries and placed into the sample chamber of a Prometheus NT.48 differential scanning fluorimeter (NanoTemper Technologies, Inc, Munich, Germany). Samples were subjected to a time-dependent temperature gradient over 20–75°C at a rate of 1°C/min. Fluorescence emission at 330 nm and 350 mn (excitation wavelength, 295 nm) was recorded at seven second intervals. The transition temperature for thermal denaturation (Tm) is defined as the temperature at the maximum the first derivative of the ratio of fluorescence emission at 350 and 330 nm (F350/F330) as determined by a polynomial fit to the temperature-fluorescence ratio curve implemented in the manufacturer's software.

(*Flock et al., 2015*) point to a common mechanism of GPCR-mediated activation that is largely consequent on binding the C-terminus and displacement of the α5 helix of Gα. In its interaction with GPCRs, as exemplified by the β2AR:Gs complex and the rhodopsin:transducin C-terminal peptide complex (*Rasmussen et al., 2011*; *Scheerer et al., 2008*), the C-terminus of Gα is accommodated by an extensive interface within a cavity formed by the trans-membrane helices of the receptor. A more limited contact surface is formed by interaction of the second cytoplasmic loop of β2R with the junction between the N-terminal helix and β1 of Gαs. Together these interactions induce structural changes in the β6-α5 loop that forms part of the purine binding site, and the P-loop that enfolds the nucleotide phosphates, thus weakening interactions between GDP and the Ras domain of Ga (*Rasmussen et al., 2011*; *Dror et al., 2015*). As observed also by DEER spectroscopy in Gi:rhodopsin (*Van Eps et al., 2011*), the interface between the helical and Ras domains is destabilized, allowing the domains to separate, providing an exit path for the nucleotide. Accordingly, HDX studies of β2AR:Gs showed strong protection of the C-terminus of Gαs (*Chung et al., 2011*). Interaction with the receptor induces strong deprotection of the P-loop and the preceding the β1 strand as well as the purine binding β6-α5 and β5-αG loops. Secondary structure elements at the interface of the Ras and helical domains, including α1 and αF, are accessible to HDX. The crystal structure of β2AR:Gs shows that Switch II retains its interactions with Gβγ as does the N-terminus of Gα. Consequently, the heterotrimer remains intact. While, in the latter complex, Gαs appears to be relatively well-ordered, earlier NMR studies of nucleotide-free [15]N-labeled transducin α bound as a complex with Gβγ to detergent-solubilized, light-activated rhodopsin showed severe broadening of amide resonances, suggestive of structure that is dynamic in the millisecond-microsecond time

scale (*Abdulaev et al., 2006*). Crystal structures may not exhibit the range of dynamic motion that is accessible to the Gαi1:GPCR complex in a lipid environment.

Ric-8A activation of free Gαi1•GDP, like GPCR activation of Gβγ-bound Gα•GDP, destabilizes the nucleotide-binding site of Gαi1 and induces domain separation, as demonstrated previously by DEER spectroscopy (*Van Eps et al., 2015*). However, the HDX data suggest that the extent of structural perturbation induced in Gαi1 by Ric-8A is at least comparable to, and possibly more extensive than those resulting from the action of the β₂AR on the Gαs, as revealed by HDX studies of the β₂AR:Gs complex (*Chung et al., 2011*). Within the Ras domain we see evidence of secondary structure conformational changes in a significant portion of the parallel β sheet scaffold that abuts and supports the nucleotide-binding site (*Figures 4a* and *5*), encompassing the hydrogen-bonded network from β1 to β4 and β5. The nucleotide phosphate-binding P loop and the β5-αG loop - which harbors the canonical NPXY motif responsible for guanine base recognition in the Ras superfamily – are strongly deprotected. Modest deprotection is observed at the β6-α5 loop that forms van der Waals contacts with the purine ring. The deprotected region extends to the interface between the Ras and helical domains (*Figure 6*) consistent with structural destabilization that facilitates domain separation, as discussed in detail above.

The question arises whether the changes in HDX associated with Ric-8A binding and nucleotide release are induced by the interaction of Ric-8A with Gαi1, or simply characteristic of the nucleotide-free state of Gαi1. Heteronuclear ¹H-¹⁵N HSQC NMR spectra of both nucleotide-free transducin α (Gαt) (*Abdulaev et al., 2006*) and Gαi1 (*Goricanec et al., 2016*) show these α subunits to be only slightly more dynamic than their GDP-bound counterparts, the former bound to Gβγ, as assessed by the broadening of amide resonances. In contrast, as described above, NMR spectra of rhodopsin: Gαt (*Abdulaev et al., 2006*) and Gαi1:Ric-8A (*Thomas et al., 2011*) show considerable line broadening indicative of dynamics in the millisecond-microsecond range. Thus, we would conclude that the substantial deprotection observed for many peptide segments of Gαi1 is largely due to structural changes induced by Ric-8A, which forms a stable, but dynamic complex with nucleotide-free Gαi1.

The mechanism by which Ric-8A induces structural changes in Gαi1 appears to differ from that used by GPCRs. While the HDX results clearly show that Ric-8A contacts the C-terminus of Gαi1•GDP, Switch II, and possibly also Switch I are also significant attachment sites. These two switches bind the GTP γ-phosphate and the Mg²⁺ cofactor in the activated state of Gα. Involvement of these regions in Ric-8A binding is supported by the biochemical and spectroscopic evidence cited above (*Thomas et al., 2011*; *Nagai et al., 2010*; *Van Eps et al., 2015*). Putative contacts with Switch II explain the very weak affinity of Ric-8A for Gα•GTP (*Tall et al., 2003*). Stabilized by backbone amide hydrogen bonds to the GTP γ-phosphate (*Sprang et al., 2007*), Switch II is well ordered in crystal structures of activated Gα proteins. Binding to Ric-8A would likely require the disruption of this energetically favorable interaction. For the same reason, Gβγ, which forms an extensive interface with Switch II, also binds weakly to Gα•GTP (*Higashijima et al., 1987b*). In contrast, Switch II is disordered in crystal structures of Gαi1•GDP (*Mixon et al., 1995*) and in solution (*Van Eps et al., 2006*). Hence, binding of Ric-8A or Gβγ to Gαi1•GDP is facilitated because Switch II is not otherwise engaged in strong interactions. Sharing a common Switch II binding site, Ric-8A and Gβγ bind competitively to Gα•GDP (*Tall et al., 2003*). However, it is likely that they also recognize different conformations of Switch II, since Ric-8A has GEF activity, while Gβγ inhibits GDP release. In aggregate, the HDX data presented here permit the inference that Ric-8A forms a large interaction surface with Gαi1. Clasping the C-terminus of Gα, and buttressed by Switch II, interaction with Ric-8A results in protection of the Gα N-terminus, parts of the α3 and α4 helices and β6 of the Ras domain, regions within the helical domain and its interface with the Ras domain (*Figure 4a*). We suggest that an extensive Ric-8A:Gαi1 interface confers stability to the dynamic and structurally heterogeneous nucleotide-free Gα. In this respect the mechanism of Ric-8A as a GEF is consistent also with its role as a Gα chaperone (*Chan et al., 2013*), in which a partially unstructured state of the Gα substrate is an intermediate in an enzyme-mediated catalytic event. Thus, the efficiency of Ric-8A-catalyzed GEF activity is limited by the rate at which Gα is able to make the transition to a partly unfolded state during GDP release and is restored to a fully folded state upon GTP binding. It is therefore not surprising that, in comparison to GPCRs, Ric-8A is a relatively inefficient GEF.

## Materials and methods

### Protein expression and purification

WT Ric-8A (1-491) was subcloned into the pET-28a vector for expression as an N-terminally hexahistidine tagged protein (*Thomas et al., 2011*). Mutants of Ric-8A were generated from the WT plasmid by user of the QuikChange II XL kit (Agilent Technologies, Santa Clara, CA). Rat Ric-8A (1-491) was expressed in *Escherichia coli* BL21 (DE3)-RIPL cells (Agilent Technologies, Santa Clara, CA) in LB media containing kanamycin (100 mg/L) and induced with 50 µM isopropyl β-D-thiogalactopyranoside (IPTG) at 20°C. After overnight growth at 20°C, cells were lysed using an EmulsiFlex-C56 cell disruptor (Avestin, Inc. Ottawa, Canada) at 4°C in lysis buffer (50 mM Tris, pH 8.0, 250 mM NaCl, 2 mM β-mercaptoethanol and 2 mM phenylmethylsulfonyl fluoride [PMSF]). The cell lysate was clarified by centrifugation and loaded onto a column containing 5 ml of Profinity IMAC resin (Bio-Rad, Hercules, CA). After extensive washing with lysis buffer, proteins were eluted from the resin with lysis buffer containing 300 mM imidazole and dialyzed against a low ionic strength buffer (50 mM Tris, pH 8.0 and 2 mM β-mercaptoethanol). The dialysate was loaded onto a HiTrap Q Sepharose FF column (GE Healthcare) and eluted with a 0–500 mM NaCl gradient on an AKTA Pure FPLC system (GE Healthcare). Pure Ric-8A 1–491 eluted from the Q-column at 165–175 mM NaCl.

The W258A mutant of rat Gαi1, encoded in a Gateway pDEST15 vector (Thermo Fisher Scientific, Waltham, MA) (*Thomas et al., 2011*), was expressed as a tobacco etch virus protease (TEV)-cleavable, N-terminal glutathione-S-transferase (GST) fusion protein and purified as described (*Thomas et al., 2008*). Nucleotide-free complex of Gαi1 and Ric-8A were prepared, with slight modifications, as described (*Thomas et al., 2011*). Ric-8A was mixed with Gαi1•GDP (1:1.5 molar ratio, both in 50 mM TRIS pH 8.0, 150 mM NaCl and 2–5 mM DTT) in a 500 µl volume and incubated at 4°C for two hours. A 50 ml slurry of agarose resin bearing immobilized alkaline phosphatase (Sigma-Aldrich, St Louis, MO) was added to the reaction mixture to degrade the released GDP and centrifuged to remove the resin. The reaction mixture was subjected to size exclusion chromatography on a Sephadex200/G75 tandem column (GE Healthcare, Chicago, IL) equilibrated in 50 mM TRIS pH 8.0, 150 mM NaCl and 2 mM DTT. The fraction corresponding to the nucleotide-free Ric-8A:Gαi1 complex were pooled and concentrated to 10–12 mg/ml using Centricon concentrators (Millipore, Darmstadt, Germany).

### Hydrogen Deuterium exchange mass spectrometry

HDX data collection was performed on an Agilent 1290 ultrahigh pressure (UPLC) series chromatography instrument coupled to a 6538 UHD Q-TOF mass spectrometer (Agilent Technologies) in the positive mode. For non-deuterated control experiments, 10 µl of protein sample (Gαi1•GDP, Gαi1•GTPγS, Ric-8A or Gαi1:Ric-8A, at concentrations of approximately 10 mg/ml) was diluted 10 fold by adding $H_2O$-based reaction buffer. Sample was quenched with a dilute formic acid solution (0.1%) at a ratio of 1:3 to achieve a final pH of 2.5. Quenched samples were digested for 2 min by addition of 15 µl of pepsin (1 mg/ml from porcine gastric mucosa, Sigma) to 60 µl of quenched protein. Quenching and digestion procedures for all samples were performed on ice.

For HDX experiments, reactions were initiated by adding 10 µl of protein or protein complex at 10 mg/ml (Gαi1•GDP, Gαi1•GTPγS, Ric-8A or Gαi1:Ric-8A) to 90 µl of $D_2O$ based reaction buffer. Proteins were incubated at 25°C and samples were removed after fixed intervals. For experiments comparing exchange rates of Gαi1•GDP vs Gαi1:Ric_8A and Ric_8A vs Ric_8A:Gαi1, samples were analyzed after 30 s, 1m, 3m, 5m, 10m, 30m, 1 hr and 5 hr after incubation in $D_2O$ buffer. For experiments comparing exchange rates of Gαi1•GDP vs Gαi1•GTPγS, time points were taken at 10 s, 10 m, 30 m, 1 hr and 5 hr. Peptides were separated using an Onyx Monolithic 100 mm X 2 mm C18 column (Phenomenix, Torrance, CA). The gradient conditions were: 3%B; 0–5.0 min, 3–23%B; 5–5.75, 98%B, 5.75–6.9 min, 98%, 6.9–7 min, 7–7.5, 98–3%. Solvent A: 0.1% formic acid (FA, Sigma-Aldrich) in water (Honeywell/Burdick and Jackson, Morris Plains, NJ) and solvent B: 0.1% FA in acetonitrile (Burdick and Jackson). Column temperature was set at 1°C. Solvents and containers were stored on ice. Electrospray conditions for MS mode were as follows: drying gas flow, 12.0 L/min at 350°C, nebulizer 60 pounds per square inch, capillary 3500V, fragmentor 120V. The scan range was 50 to 1700 m/z. Data acquisition and spectral analysis were performed using MassHunter (Qualitative Analysis version B.04.00, Agilent Technologies). Each experiment was repeated three times.

Using Agilent Mass Hunter B.04.00 and Bio-Confirm, unique peptides were identified in solo protein samples (Gαi1•GDP, Ric-8A) and for the complex (Gαi1:Ric-8A). The data presented here is based on peptides that were unambiguously assigned a unique sequence position based on unique mass (±10 ppm) and MS/MS sequencing (100 ppm for fragmentation data. The list of identified unique peptides containing sequence, peptide m/z, charge and retention time was used as input for HDExaminer Version 1.3.0 beta6 (Sierra Analytics, Modesto, CA) to calculate the centroid of the isotope distribution of the deuterated peptides at different time points. HDExaminer was also used to determine the peptide quality by comparing the isotopic distribution of observed and predicted peptides. The standard deviation reported is the difference in deuterium uptake between three complete technical replicates (*Figure 5a* and *Figure 6a*) in which three samples of the same preparation were were subjected to experimental analysis.

Peptide sequence tags were generated by MS-MS data acquisition, with a scan range of 50–1700 m/z (auto MS-MS)) with an isolation width of 4 m/z and an acquisition rate of 1 spectrum/s. Four fixed collision energy voltage (20V, 25V,30V and 40V) and linear gradient voltages were applied in auto MS-MS mode. MS-MS data was analyzed and mapped to confirm the corresponding sequence using Bruker Daltonics Bio Tools 3.2 (Bruker Daltonics Inc, Billerica, MA). In addition, Peptide shaker software (Compomics, Gent, Belgium) was used to confirm assingments (*Vaudel et al., 2015*). For both Bruker and Peptide shaker analysis, raw MS-MS data files were converted to mgf format before data mapping. After the completion of data analysis, heat maps (*Figures 1–3*) were generated using the web-based application, MS tools (*Kavan et al., 2011*).

Back-exchange during chromatography was estimated from a fully deuterated protein. The 24 hr time point was denoted as 100% ($m_{100}$) exchange. The non-deuterated control was denoted as 0% ($m_{0\%}$) exchange. Percentage deuteration for each peptide at different time points was calculated by following formula:

$$\mathrm{DP} = \left(\frac{m_t - m_{0\%}}{m_{100\%} - m_{0\%}}\right) \bullet 100$$

where $m_t$ is peptide mass after incubation in deuterated buffer at time $t$.

To represent the extent of HDX on the cartoon representations of Gαi1 and Ric-8A (*Figure 4–7*) generated in PyMOL (*Schrödinger, 2012*), each Cα atom was assigned a percent deuteration value $D_i$,

$$D_i = \frac{\sum_{j=1}^{N^i} w_j^i < DP_j^i >}{\sum_{j=1}^{N_i} w_j^i}$$

where $< DP_j^i >$ is the average % deuterium uptake over the first two time points (30 s and 60 s) for the $j^{th}$ of the $N^i$ peptides that contain residue $i$, and $w_j^i$ is a weighting factor for $DP_j^i$, equal to the lesser of: $1/L_j^i$, where $L_j^i$ is the length of the peptide, or 0.2. The color assigned to each residue is scaled to the value of $D_i$ according to the color key shown in *Figure 4*.

## Assay of Ric-8A guanine nucleotide exchange activity

Ric-8A-catalyzed exchange of GTPγS for GDP at Gαi1 (blue histogram bars in *Figure 7b*) was followed by monitoring the change in intrinsic fluorescence of Gαi1 at 345 nm upon exchange of GDP with GTPγS (*Higashijima et al., 1987a*). Gαi1•GDP in exchange buffer composed of 50 mM HEPES, pH 8.0, 150 mM NaCl, 10 mM MgCl$_2$, and 1 mM TCEP in a reaction volume of 500 μl was allowed to equilibrate for 5 min at 25°C in a quartz fluorescence cuvette with stirring. Ric-8A samples were equilibrated separately at 25°C. GTPγS in exchange buffer was added to the reaction mixture to a final concentration of 20 μM in the absence or presence of Ric-8A, and the increase in fluorescence at 345 nm was monitored upon excitation at 295 nm. Fluorescence measurements were conducted using an LS55 fluorescence Spectrometer (PerkinElmer Life Sciences, Waltham MA). The excitation and emission slit widths were set at 2.5 nm. All exciting light was eliminated by use of a 290 nm cutoff filter positioned in front of the emission photomultiplier.

To determine the rate of exchange of Ric-8A for GTPγS at Gαi1:Ric-8A (orange histogram bars in *Figure 7b*), Ric-8A was added to the cuvette containing Gαi1:GDP after a 5 min equilibration at

25°C. The reaction mixture was incubated for another 5 min before addition of GTPγS in exchange buffer to a final concentration of 20 μM.

10 min time courses of GTPγS binding were recorded and fit to a single exponential equation using SigmaPlot (Systat Software Inc., San Jose, CA):

$$\Delta F_t = \Delta F_{max}\left(1 - e^{-kt}\right)$$

Where $\Delta F_t$ is the increase in fluorescence relative to the baseline at time $t$. $\Delta F_{max}$ is the maximum change in in tryptophan fluorescence determined by extrapolation and k is the rate constant, in $min^{-1}$ for the first-order change in fluorescence with time.

The initial velocity of nucleotide exchange was calculated by taking the first derivative of this equation:

$$v_t = \frac{dF_t}{dt} = -k\,\Delta F_{max}\left(1 - e^{-kt}\right)$$

Where, at $t = 0$,

$$v_0 = \frac{dF_t}{dt} = k\,\Delta F_{max}$$

For each experiment, the molar rate of nucleotide exchange was calculated using the ratio $\Delta F_{max}/[G\alpha i1\bullet GDP]_0$ where $[G\alpha i1\bullet GDP]_0$ is the initial concentration of $[G\alpha i1\bullet GDP]$ in the reaction as determined from its computed extinction coefficient at 280 nm. The relative activity of each Ric-8A mutant was computed as the ratio of its GEF activity ($v_0$) to that of WT Ric-8A. For each Ric-8A mutant, relative activities were determined for each of three samples derived from the same stock solution of protein and the average relative activity and standard deviation computed. Assays were conducted over a period of several days, using the same stock of Gαi1 and WT Ric-8A, and the activity of WT Ric-8A re-determined each day from a single sample.

P-values associated with the difference between the GEF activities of mutant *versus* WT Ric-8A were conducted using a two-tailed Student's t test based on the mean and variances of three determinations for the activities of each mutant and eight determinations of the activity of WT Ric-8A

## Acknowledgements

This work was supported by the US National Institutes of health (NIH) (grants R01-GM105993 to S.R. S. The mass spectrometry facility at Montana State University receives funding from the Murdock Charitable Trust and NIH P20RR02437. The CBSD Macromolecular X-ray Diffraction Core (MDXC), whose protein expression facilities were used to conduct this research at the University of Montana receives support from NIH P20-GM103546. We are grateful to Dr. Tung-Chung Mou of the MDXC, and Drs. Charles Heffern and Ali Raja of NanoTember Technologies, Inc. for assistance with differential scanning fluorescence assays, Cindee-Yates Hansen and Marlene Woldsted for excellent technical assistance, Dr. Jonathan Hilmer for assistance with mass spectrometry, Navid Moavhed for helpful discussions on HDX and Dr. Gregory Tall for his mechanistic insights into the function of Ric-8A in cells. We thank Angela Patterson for creating *Video 1*.

## Additional information

### Funding

| Funder | Grant reference number | Author |
| --- | --- | --- |
| M.J. Murdock Charitable Trust | | Brian Bothner |
| National Institutes of Health | P20-RR002437 | Brian Bothner |
| National Institutes of Health | P20-GM103596 | Stephen R Sprang |
| National Institutes of Health | R01-GM105993 | Stephen R Sprang |

The funders had no role in study design, data collection and interpretation, or the decision to submit the work for publication.

## Author contributions
RK, BZ, Acquisition of data, Analysis and interpretation of data; CJT, prepared and purified proteins and protein complexes for HDX analysis, Contributed unpublished essential data or reagents; BB, SRS, Conception and design, Analysis and interpretation of data, Drafting or revising the article

## Author ORCIDs
Stephen R Sprang, http://orcid.org/0000-0001-6307-6166

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
