## [Decision Letter]

Thank you for submitting your article "The cytoplasmic nucleotide exchange factor Ric-8 induces long-range secondary structure changes in Gα" for consideration by *eLife*. Your article has been reviewed by three peer reviewers, and the evaluation has been overseen by John Kuriyan as the Reviewing Editor and Senior Editor. The reviewers have opted to remain anonymous.

The reviewers have discussed the reviews with one another and the Editor has drafted this decision to help you prepare a revised submission.

The canonical mechanism for the activation of heterotrimeric G-proteins is through membrane-bound GPCRs. Ric-8A is cytoplasmic molecular chaperone for the G-α proteins, and also has non-canonical guanine nucleotide exchange (GEF) activity for the G proteins (although much weaker than that of GPCRs). Activation of the G-proteins results from stabilization of G-α in the nucleotide free state, facilitating GTP binding. In this manuscript, Kant et al. describe the interaction between Ric-8A and G-α, and propose a mechanism that underlies the accelerated exchange activity. From past work, although much functional evidence concerning this interaction had been resolved, very little knowledge had been gained on the structural constraints on G-α that underlies the exchange activity of Ric-8A. This dearth of structural information is due to the relatively dynamic nature of the interaction.

In this interesting study the investigators utilize deuterium exchange mass spectrometry (which we shall refer to simply as HDX) to both follow the conformational changes in G-α upon Ric-8A binding, as well as to establish the footprint of the protein-protein interface. By monitoring changes in rates of deuteration, one can generate a map of which regions are involved in the interface (as these are less exposed to solvent) or that are linked to binding allosterically. Interpretation of the data depends on having good atomic models for the two interacting proteins and a solid molecular understanding of roles and dynamic range of their flexible/functional elements. In this study, however, only G-α is known at that level of structural detail because there is no atomic structure of Ric-8A. Thus, a weakness of this work is that although multiple sites of interaction are identified, and the central hypothesis that Ric-8A stabilizes a poorly ordered nucleotide G protein subunit (consistent with chaperone function) is well supported, one still cannot construct a reasonable model of how the C-terminus and switch regions of G α interact with Ric-8A, in particular because the key region of Ric8A itself is thought to be intrinsically disordered. Nevertheless, despite this limitation, the work is of high technical quality, and the data are interpreted in a reasonable light. On balance, given that Ric-8A has proven to be a tough structural target and is thought to be highly dynamic on its own, this kind of study may be the highest resolution information, when used in combination with previously published DEER studies, that can be obtained for this biologically important and interesting complex.

More specifically, this study shows that Ric-8A shares some key properties with GPCRs in how they interact with G-α, in that the C-terminal helix of the G protein is likely a key interaction site; motions of this helix are well known to be structurally coupled to regions involved in nucleotide binding. But there are some key differences too. The first is that there are broad increases in flexibility across more of the Ras-like domain upon complex formation (a loosening of the tertiary structure) and interactions also appear to be formed between Ric-8A and two of the switch regions of the α subunit. The authors go on to show that a region predicted to be disordered at the C-terminus of Ric-8A is highly protected from exchange upon complex formation. The authors hypothesize that this is an important G protein binding site, an idea they support via site-directed mutagenesis and truncation. These Ric-8A variants display significantly reduced nucleotide exchange on Gαi.

The paper presents observations that the reviewers find to be novel and fascinating, and will be of interest not only to specialists in GPCR function but also to scientists interested in the broader and diverse class of guanine-binding proteins. The manuscript is well written and the experimental design is rigorous. The data are of high quality and support the conclusions. The authors are appropriately cautious in their interpretation. Thus, despite the reservations expressed about some aspects of the work, the reviewers have concluded that the paper represents a substantial advance in terms of furthering our understanding of G-proteins, and is therefore suitable for publication in *eLife*.

The reviewers have not identified any fundamental weakness in the paper that must be corrected prior to publication. However, they have raised a number of issues that the authors are encouraged to address in a revised manuscript.

General points raised by the reviewers:

1) One minor concern is the validity of the Ric-8A model – can this be validated by CD or other biophysical measures? Also, the biophysical analysis is augmented by some biochemical validation, but there is no follow up in a biological setting. What are the physiological consequences of mutating Ric-8A to block interaction with Gα in cells? Are they distinct from those of a Ric-8A knock out (that would suggest that Ric-8A has other targets to be identified)? This could be done in *C. elegans, Drosophila*, or cultured cells. We recognize that such experiments are likely to be outside the scope of this work, but if the authors have any additional insights concerning these points it would be helpful to include these in the revised manuscript.

2) The authors might want to soften their tone slightly on the role of the C-terminus and the 5-6 loop on nucleotide exchange in light of the recent crystal structure of the adenosine A2A receptor bound to a mutant G-α subunit that contains only the Ras domain (Carpenter et al., Nature, 2016). In this structure the G-α subunit remains bound to GDP, despite engaging the C-terminus in a similar manner as in the β-adrenergic receptor-G protein complex. More importantly, not only was the 5-6 loop disordered (or not modelled) in the structure but the N-terminus of G-α was absent. In light of these recent data the authors should modify their discussion.

3) One of the reviewers suggests that it might be worth discussing in the this paper the NMR results of the late Kevin Ridge, who looked at the structure of transducin in complex with rhodopsin. This reviewer recalls that the Ridge lab generated results about the partially unfolded state of the Ras-like domain that bear some resemblance to the results of this study. With this in mind, the β_2_AR-Gs complex study seems like a bit of an outlier in how little unfolding there is in the α subunit in its nucleotide free state. It also argues that G-β-γ isn't necessarily responsible for the stability of Gs in the β_2_AR over that observed for Gi in the Ric-8A complex. Or does this just reflect that there will be a spectrum of relative unfolding given the identity of the GEF and the α subunit?

4) The authors haven't really discussed the difference in the HDX kinetics that are Ric8-dependent, as opposed to the kinetics of the nucleotide-free form of G-α. Could an attempt be made to distinguish the deuterium exchange on G-α from the ric8 protein-protein interaction from the difference in exchange rates simply because the G protein is nucleotide-free. Would it be possible, for example, to study the kinetics of ric8 interaction with G-α by DXMS? Would it also be possible to study the kinetics of exchange in the G-α-Ric8 complex following high concentrations of GDP? A similar argument could be made about the previously reported EPR study on the G-α-Ric8 complex (van Eps et al. 2015). This is relevant since it is not quite clear whether Ric8 truly stabilizes the 'open' conformation of the Gα, or simply the nucleotide-free form, which happens to be in the 'open' conformation.

5) It is not really clear whether Ric8 is a physiological GEF of G-αs or whether they serve only as chaperones. While there is cellular evidence of the latter there really isn't any satisfying evidence of the former. Therefore, it is suggested that Ric8 might really be referred to as having or displaying GEF activity but not necessarily be referred to as a GEF, or a physiologically relevant GEF.

Specific points raised by the reviewers:

1) In subsection “Secondary structure plasticity of Gαi1 in the Gαi1:Ric-8A complex” the authors refer to the rapid exchange kinetics of the b6 strand, with the Ric8A-bound form initially exchanging slightly slower. Looking at the data in Figure 2 it hard to see what differences there are. More importantly why should the exchange kinetics of this region be so fast in the GDP-bound form and apparently insignificantly different when bound to Ric8 and nucleotide-free?

2) Paragraph two of subsection “GDP **→** GTP exchange reverses Ric-8A – induced conformational changes” should be clarified. I believe that the authors mean that the N-terminal region of the b2-strand and C-terminal region of swII are strongly protected. Also Asp 200 is likely too far to directly be involved in γ phosphate binding and is probably interacting though a water.

3) The legend in Figure 4 is not clear. It is not really clear what the blue and orange bars really represent, other than that they are relative exchange rates.

4) We ask that the Supplemental Figure 2, Figure 3 and Figure 4 be moved to the main set of figures. These figures are closer to the primary data than are the exchange differences mapped on to the structure (current Figure 1), and will be helpful in assessing the significance of the changes shown in the current Figure 1. Given that there are no page constraints in *eLife*, any figures that are central to the argument should be in the main body of the paper, not in supplemental figures.

5) In the first paragraph of the paper, where it is stated that Ric-8A is "the best characterized of the family of cytosolic guanine nucleotide exchange factors", the language should be changed. This statement is true only for heterotrimeric G-proteins, since cytosolic exchange factors for Ras and other small G-proteins are very well understood.

6) In the paragraph that begins: "Ric-8A forms a Michaelis complex with G-α.GDP.…", the writing implies that there is a preferential difference in the direction of nucleotide exchange (GDP replaced by GTP). For Ras-specific exchange factors, there is no such difference, and the direction is set simply by the cellular concentration of GTP vs GDP. The authors should clarify whether there is a known preference for the direction of exchange, or, if this is not known, then the language should be adjusted.

7) There are several typographic errors and problems with the symbol font throughout the manuscript, which should be reviewed carefully.

---

## [Author Response]

General points raised by the reviewers:

1) One minor concern is the validity of the Ric-8A model – can this be validated by CD or other biophysical measures? Also, the biophysical analysis is augmented by some biochemical validation, but there is no follow up in a biological setting. What are the physiological consequences of mutating Ric-8A to block interaction with Gα in cells? Are they distinct from those of a Ric-8A knock out (that would suggest that Ric-8A has other targets to be identified)? This could be done in C. elegans, Drosophila, or cultured cells. We recognize that such experiments are likely to be outside the scope of this work, but if the authors have any additional insights concerning these points it would be helpful to include these in the revised manuscript.

Circular dichroic spectroscopic spectra have been measured by Ric-8A, which are indicative of an all a-helical structure. We have cited the relevant studies in the Introduction. With regard to the second point, we are not aware of studies that in which mutations that specifically abrogate Ric-8A:Gα interactions have been made. However, full knock-out of Ric-8A in mouse results in early embryonic lethality. This study is cited in the Introduction

2) The authors might want to soften their tone slightly on the role of the C-terminus and the 5-6 loop on nucleotide exchange in light of the recent crystal structure of the adenosine A2A receptor bound to a mutant G-α subunit that contains only the Ras domain (Carpenter et al., Nature, 2016). In this structure the G-α subunit remains bound to GDP, despite engaging the C-terminus in a similar manner as in the β-adrenergic receptor-G protein complex. More importantly, not only was the 5-6 loop disordered (or not modelled) in the structure but the N-terminus of G-α was absent. In light of these recent data the authors should modify their discussion.

The reviewers suggest that we soften our assertion of the role of Ric-8A induced perturbation of β6-α5 in the release of GDP. Indeed, we observe deprotection of this loop region, which forms van der Waals contacts with the purine ring, but do not claim structural changes in this peptide are crucial to nucleotide release, but do point out that the β6-α5 loop is substantially displaced in the interaction of the Gαs with β2AR as observed in the crystal structure of the Gs:β2R complex. (there is other evidence from to suggest the role of β6-α5 loop, but beyond the scope of the discussion in this manuscript. The HDX data do not support a detailed mechanistic hypothesis regarding the role of this structural element).

One reviewer notes that in the recently published structure of the A2A receptor bound to a modified Ras domain of Gαs, GDP remains bound despite the presumptive disordering of residues in the b6-a5 loop (Carpenter, et al., Nature, 2016). We would contend that relevance of this finding is open to question. The Gαs Ras domain used to form the complex with A2AR (Mini-Gs395) contains eight mutations, including a deletion within switch III and is N-terminally truncated (Carpenter and Tate, Protein Eng Des Sel, 29:1-11, 2016). The mutations were designed, and subsequently shown, to increase the thermostability of the domain. However, these mutations also compromise nucleotide release, to the extent that the A2AR:mini-GS395 complex does not dissociate in the presence of GTP. The crystal structure of the complex shows that the contacts between the mini-Ras domain and GDP are largely intact in the complex with A2AR. Although residues in the Gαs β6-α5 loop are not modeled (coordinates for a three-residue sequence in the loop are not present in the PDB file submitted with the paper), inspection of the 2mFo-DFc map show that the loop is well ordered and residues within the loop make contact with the purine ring in a fashion similar to that observed in typical Gα•GDP complexes. For these reasons, we do not feel that the structure of the A2AR-miniGs395 complex informs our understanding of the HDX results presented in the manuscript.

3) One of the reviewers suggests that it might be worth discussing in the this paper the NMR results of the late Kevin Ridge, who looked at the structure of transducin in complex with rhodopsin. This reviewer recalls that the Ridge lab generated results about the partially unfolded state of the Ras-like domain that bear some resemblance to the results of this study. With this in mind, the β_2_AR-Gs complex study seems like a bit of an outlier in how little unfolding there is in the α subunit in its nucleotide free state. It also argues that G-β-γ isn't necessarily responsible for the stability of Gs in the β_2_AR over that observed for Gi in the Ric-8A complex. Or does this just reflect that there will be a spectrum of relative unfolding given the identity of the GEF and the α subunit?

One of the reviewers suggests that we refer to the rhodopsin:Gt complex by Kevin Ridge and his colleagues by HSQC NMR conducted by Kevin Ridge and his colleagues (Abdulaev, et al., Biochemistry, 2006). We are grateful for this suggestion, and have followed it (see Discussion section). However, as shown in a survey conducted by Montelione and colleagues (JACS, 127:16505, 2005), high-resolution crystal structures have been determined of proteins that exhibit poor (broadened) HSQC spectra. The reviewer’s argument for a range in structural dynamics for various GPCR or Ric-8A complexes with nucleotide-free Gα is reasonable, but crystal structures may not reveal the full accessible range of dynamics.

4) The authors haven't really discussed the difference in the HDX kinetics that are Ric8-dependent, as opposed to the kinetics of the nucleotide-free form of G-α. Could an attempt be made to distinguish the deuterium exchange on G-α from the ric8 protein-protein interaction from the difference in exchange rates simply because the G protein is nucleotide-free. Would it be possible, for example, to study the kinetics of ric8 interaction with G-α by DXMS? Would it also be possible to study the kinetics of exchange in the G-α-Ric8 complex following high concentrations of GDP? A similar argument could be made about the previously reported EPR study on the G-α-Ric8 complex (van Eps et al. 2015). This is relevant since it is not quite clear whether Ric8 truly stabilizes the 'open' conformation of the Gα, or simply the nucleotide-free form, which happens to be in the 'open' conformation.

This is an excellent question, but one that we have not addressed by HDX, in part due to the instability of nucleotide-free Gαi1 under the conditions used for the experiments shown in Figure 1–Figure 3. However, the two HSQC studies demonstrate that the nucleotide-free “apo” structures show considerably less amide broadening than the complexes with GPCRs. On this basis, we would suggest that the major changes in HDX protection are indeed due to stabilization of a dynamic state by Ric-8A. We are conducting HSQC studies with Gαi1 and Ric-8A in collaboration with Franz Hagn’s group, and we would say (in confidence) that the preliminary data bear this out.

5) It is not really clear whether Ric8 is a physiological GEF of G-alphas or whether they serve only as chaperones. While there is cellular evidence of the latter there really isn't any satisfying evidence of the former. Therefore, it is suggested that Ric8 might really be referred to as having or displaying GEF activity but not necessarily be referred to as a GEF, or a physiologically relevant GEF.

We agree that the evidence is suggestive but not conclusive. We have made changes throughout the Introduction (including the title) to address this point.

Specific points raised by the reviewers:

1) In subsection “Secondary structure plasticity of Gαi1 in the Gαi1:Ric-8A complex” the authors refer to the rapid exchange kinetics of the b6 strand, with the Ric8A-bound form initially exchanging slightly slower. Looking at the data in Figure 2 it hard to see what differences there are. More importantly why should the exchange kinetics of this region be so fast in the GDP-bound form and apparently insignificantly different when bound to Ric8 and nucleotide-free?

The reviewer is correct in that the difference between the GDP and Ric-8A-bound complexes with respect to HDX is small. Seven overlapping peptides report on this region. Indeed, as Figure 5 shows, β6 is slightly protected by Ric-8A. On re-inspection of the uptake curves (Source Data Table 1), it appears that only one shows rapid deprotection within the first 60 seconds. The others show deuteration in the range of 55%-60% for Gαi1•GDP and 45%-55% for Gαi1:Ric-8A. The exceptional peptide (315-323) is, unfortunately, the one depicted in Figure 5, and we suspect that the% deuteration measured for this peptide at the 30 second time point is erroneous, since none of the other peptides that overlap in this region show a similar value. Therefore we have replaced this panel with the uptake curve of a representative peptide (for the peptide spanning 311-323). The Discussion section has been revised to reflect the slight protection that Ric-8A confers on the region extending from α4 to β6.

*2) Paragraph two of subsection “GDP*
***→***
*GTP exchange reverses Ric-8A – induced conformational changes” should be clarified. I believe that the authors mean that the N-terminal region of the b2-strand and C-terminal region of swII are strongly protected. Also Asp 200 is likely too far to directly be involved in γ phosphate binding and is probably interacting though a water.*

The confusion is due to a typographical error. The revised text states that the C-terminus of β3 and switch II are strongly protected. We further note that Asp 200 does not directly contact the γ phosphate of GTP, but rather, forms a water-mediated contact with the magnesium cofactor.

3) The legend in Figure 4 is not clear. It is not really clear what the blue and orange bars really represent, other than that they are relative exchange rates.

The legend to Figure 7 (formerly Figure 4) has been revised to better describe the data represented by orange and blue bars. The Materials and methods section now indicates the experiments that are represented in the histogram, and the Results/Discussion section now reminds the reader that incorporation of GTPγS into the Gαi1 in the complex with Ric-8A proceeds with the dissociation of Ric-8A.

4) We ask that the Supplemental Figure 2, Figure 3 and Figure 4 be moved to the main set of figures. These figures are closer to the primary data than are the exchange differences mapped on to the structure (current Figure 1), and will be helpful in assessing the significance of the changes shown in the current Figure 1. Given that there are no page constraints in eLife, any figures that are central to the argument should be in the main body of the paper, not in supplemental figures.

Supplemental figures 2-4 have now been incorporated into the main manuscript as Figure 1–Figure 3. The figures have been modified to show the location of secondary structure elements (determined from X-ray structures for Gαi1, and predicted from JPRED4 analysis for Ric-8A). The figures also highlight the residues involved in nucleotide binding in Gαi1, and putative Gαi1 binding in Ric-8A.

5) In the first paragraph of the paper, where it is stated that Ric-8A is "the best characterized of the family of cytosolic guanine nucleotide exchange factors", the language should be changed. This statement is true only for heterotrimeric G-proteins, since cytosolic exchange factors for Ras and other small G-proteins are very well understood.

The Introduction has been amended to indicate that Ric-8A is the best characterized of the cytosolic exchange factors for heterotrimeric G protein α subunits. We also indicate that two other armadillo repeat proteins have been identified (importin-β and smgGDS) that interact in differing functional contexts with certain small Ras-like G proteins.

6) In the paragraph that begins: "Ric-8A forms a Michaelis complex with G-α.GDP.…", the writing implies that there is a preferential difference in the direction of nucleotide exchange (GDP replaced by GTP). For Ras-specific exchange factors, there is no such difference, and the direction is set simply by the cellular concentration of GTP vs GDP. The authors should clarify whether there is a known preference for the direction of exchange, or, if this is not known, then the language should be adjusted.

The GEF activity of Ric-8A is indeed unidirectional, exchanging GDP for GTP, but not the reverse. This is noted in the Introduction.

7) There are several typographic errors and problems with the symbol font throughout the manuscript, which should be reviewed carefully.

The typographical errors noted, together with others subsequently discovered, have been corrected.